# Immunoglobulin Superfamily Containing Leucine-Rich Repeat (ISLR) Serves as a Redox Sensor That Modulates Antioxidant Capacity by Suppressing Pyruvate Kinase Isozyme M2 Activity

**DOI:** 10.3390/cells13100838

**Published:** 2024-05-14

**Authors:** Tongtong Wang, Meijing Chen, Yang Su, Yuying Zhang, Chang Liu, Miaomiao Lan, Lei Li, Fan Liu, Na Li, Yingying Yu, Lei Xiong, Kun Wang, Jin Liu, Qing Xu, Yue Hu, Yuxin Jia, Yuxin Cao, Jingwen Pan, Qingyong Meng

**Affiliations:** State Key Laboratory of Agrobiotechnology, College of Biological Sciences, China Agricultural University, Yuanmingyuan West Road No. 2, Haidian District, Beijing 100193, China; vetwangtongtong@163.com (T.W.); cmj13007570425@163.com (M.C.); suyangsy@aliyun.com (Y.S.); changliu2020@163.com (C.L.); lilei1995@cau.edu.cn (L.L.); 18309856051@163.com (N.L.);

**Keywords:** oxidative stress, ISLR, redox balance, pyruvate kinase

## Abstract

Cells defend against oxidative stress by enhancing antioxidant capacity, including stress-activated metabolic alterations, but the underlying intracellular signaling mechanisms remain unclear. This paper reports that immunoglobulin superfamily containing leucine-rich repeat (ISLR) functions as a redox sensor that responds to reactive oxygen species (ROS) stimulation and modulates the antioxidant capacity by suppressing pyruvate kinase isozyme M2 (PKM2) activity. Following oxidative stress, ISLR perceives ROS stimulation through its cysteine residue 19, and rapidly degrades in the autophagy–lysosome pathway. The downregulated ISLR enhances the antioxidant capacity by promoting the tetramerization of PKM2, and then enhancing the pyruvate kinase activity, PKM2-mediated glycolysis is crucial to the ISLR-mediated antioxidant capacity. In addition, our results demonstrated that, in triple-negative breast cancer, cisplatin treatment reduced the level of ISLR, and PKM2 inhibition sensitizes tumors to cisplatin by enhancing ROS production; and argued that PKM2 inhibition can synergize with cisplatin to limit tumor growth. Our results demonstrate a molecular mechanism by which cells respond to oxidative stress and modulate the redox balance.

## 1. Introduction

Reactive oxygen species (ROS) in cells are mainly composed of hydrogen peroxide (H_2_O_2_), superoxide (O_2_^−^), and hydroxyl radicals (•OH) [1]. The balance of ROS production and elimination is critical for maintaining redox homeostasis. Excessive ROS production can lead to the oxidative damage of intracellular macromolecules, which can eventually lead to cell death [2]. In addition to promoting cellular damage and cell death, ROS are also essential for the regulation of several other cellular responses, and have long been recognized as small molecules that function as factors of signal transmission.

As signal factors, ROS participate in multiple processes, including initiating apoptotic signaling to induce cell death or mediate metabolic reprogramming, to altering the redox balance [3,4]. For example, NF-E2-related factor 2 (NRF2) is an ROS-responsive transcriptional factor. Under unstressed conditions, NRF2 is degraded through the ubiquitin–proteasome pathway in a Keap1-dependent manner. However, under oxidative stress conditions, Keap1 is oxidized and unable to bind with NRF2; then, NRF2 is released from Keap1 and regulates the transcription of various metabolic enzymes (including the *glutamate-cysteine ligase catalytic subunit* and *glutathione peroxidase*) to enhance the intracellular antioxidative capacity, therefore maintaining the redox balance [5]. Although ROS are widely recognized as a signaling factor, little is known about ROS sensing and the direct regulation of ROS-signaling molecules.

Immunoglobulin superfamily containing leucine-rich repeat (ISLR) is a member of the leucine-rich repeat and immunoglobulin family of proteins [6]. Previous reports have demonstrated that ISLR is widely expressed in the skeletal muscle, heart, thyroid, brown adipose tissue, cancer-associated fibroblasts, and various cancer cells [7,8,9,10,11]. It is proven that ISLR is a potential marker for mesenchymal stromal cells [12], and it regulates skeletal muscle regeneration by activating Wnt signaling [8]. Hara et al. showed that aging and hypoxia could induce the downregulation of ISLR [13]; as both aging and hypoxia have been widely reported to induce ROS [13,14], we speculated that ISLR might participate in ROS sensing. In addition, GEO profiles showed that the expression of *ISLR* was suppressed by *Nrf2* activation (GDS3476/1418450_at). Meanwhile, a glutathione supplement fully rescued the increased *ISLR* expression which was induced by *Nrf2* deletion (GDS2875/1418450_at) [15]. These data indicate that *ISLR* might be involved in the antioxidant process. However, it remains unclear whether *ISLR* participates in sensing ROS and regulating redox balance.

In this study, we found that ISLR is an ROS sensor. Upon oxidative stress, ISLR is rapidly degraded in the autophagy–lysosome pathway; then, the decreased ISLR level could improve the intracellular antioxidant capacity. Mechanically, we found that ISLR suppresses pyruvate kinase isozyme M2 (PKM2) tetramerization to decrease the pyruvate kinase activity, and, therefore, suppresses the antioxidant capacity. Moreover, we found that PKM2 activity is critical to the antioxidant capacity; limiting the PKM2 activity sensitizes tumors to chemotherapy. Consequently, a combined treatment of the PKM2 inhibitor and chemotherapy is a potential strategy for triple-negative breast cancer therapy.

## 2. Materials and Methods

### 2.1. Cell Culture

C3H10T1/2, HEK293T, A549, and C2C12 cell lines were purchased from the Chinese Academy of Medical Sciences and the School of Basic Medicine at Peking Union Medical College. The 4T1 cell line was provided by Dr. Zhengquan Yu (State Key Laboratory of Agrobiotechnology, China Agricultural University, Beijing, China). MEFs were isolated from C57/BL6 mice according to published protocols [16]. The 4T1 and A549 cells were cultured in RPMI-1640 (Cat#C11875500, Gibco, Waltham, MA, USA) supplemented with 10% fetal bovine serum (Cat#10099-141, Gibco) and 1% penicillin-streptomycin (Cat#15070063, Gibco) in an atmosphere with 5% CO_2_ unless otherwise indicated. HEK293T, C3H10T1/2, C2C12, and MEFs were cultured in DMEM (Cat#C11995500BT, Gibco) supplemented with 10% fetal bovine serum and 1% penicillin-streptomycin. For three-dimensional culture, 4T1 cells were seeded in 96-well nunclon sphera-treated U-shaped microplate (Cat#174925, Thermo Fisher, Waltham, MA, USA).

### 2.2. Animal Studies

All animal studies were performed according to protocols approved by the China Agricultural University Laboratory Animal Welfare and Animal Experimental Ethical Committee. All the mice were raised in pathogen-free conditions and were fed with a pathogen-free diet and water and were housed under a 12:12 h L:D photoperiod at 25 ± 1 °C. Balb/c mice were bought from Beijing Vital River Laboratory Animal Technology Co. Ltd. (Beijing, China). *Islr-*deleted mice were from Cyagen Biosciences (Guangzhou) Inc. (Beijing, China).

### 2.3. Plasmids and Transfections

Expression plasmid for *Islr*-GFP was constructed as previously described [8]. *Pkm2*-Flag, *Pkm2*-3×Flag, *Islr*, *Islr*-Flag, and their derivative mutants were subcoloned into the NheI and BamHI sites of the pCDH-CMV-MCS-EF1-copGFP-T2A-Puro vector. Point mutants of ISLR-Flag were constructed using PCR-based mutagenesis.

For overexpression of *Islr*, 8 × 10^5^ cells per well were seeded into 6-well plates, and then transfected with 2.5 μg *Islr* plasmid per well using Lipofectamine 2000 (Cat#11668019, Invitrogen, Waltham, MA, USA) according to the manufacturer’s protocol. Stable A549 cells were further selected by geneticin 1 d after transfection.

The *Islr*-deleted cell lines were generated by the CRISPR-Cas9 technology. Guide RNA targeting mouse *Islr* was cloned into a px330 vector including a Cas9-expressing cassette to construct the *Islr*-knockout plasmid. For *Islr*-deletion in 4T1 cells, 3 × 10^5^ cells per well were seeded into 12-well plates, and then transfected with 1 μg *Islr*-knockout plasmid per well using Lipofectamine 2000. Forty-eight hours after transfection, single clones were cultured in 96-well plates; gene sequencing was used to verify *Islr*-deleted cells. The sequence of guide RNA used in this study is as follows: *Islr* gRNA, GATCTCCACCGTGCCGCCCG.

### 2.4. Cell Viability Assay

CellTiter-Glo™ Luminescent Cell Viability Assay Kit (Cat#G7570, Promega, Wisconsin, WI, USA) was used to measure cell viability according to manufacturer’s instructions. In brief, 3 x 10^4^ cells were seeded in white-wall 96-well plates, 24 h after incubation; cells were treated with indicated reagents (H_2_O_2_ with or without 2-DG (Cat#HY-13966, MedChemExpress, New Jersey, NJ, USA)/Bay876 (Cat#HY-100017, MedChemExpress)) for 3 h. Then, 100 μL CellTiter-Glo^®^ reagent was added to the experimental wells, incubated at room temperature for 10 min, and then luminescence was recorded with a luminometer.

### 2.5. Live Cell Number Measurement

Cells were seeded in 6-well plates at same cell density, and then treated with different reagents 12 h later (treatments were the same as described above); cells were trypsinized and collected, and then stained with trypan blue (Cat#15250061, Gibco) and counted immediately with Thermo Fisher Countess II FL cell counter to evaluate total live cell number.

### 2.6. Intracellular ROS Levels

To measure intracellular ROS levels, cells grown in 12-well plates were washed twice with FBS-free RPMI-1640, stained with 10 μM Dichlorodihydrofluorescein diacetate (DCFH-DA) (Cat#S0033, Beyotime, Shanghai, China) and 10 μg/mL Hoechst (Cat#C1011, Beyotime) in FBS-free RPMI-1640 culture medium at 37 °C for 30 min under 5% CO_2_, and washed again with FBS-free RPMI-1640. Fluorescence was visualized using an Echo Revolve Hybrid Microscope Optical microscope (Echo, San Diego, CA, USA). The cells which have DCF fluorescence (green) and DAPI fluorescence (blue) were quantified separately using ImageJ version 1.47 software; Relative DCF fluorescence was obtained by counting DCF positive cell numbers/DAPI fluorescence cell numbers. All measurements were performed blinded to condition.

### 2.7. Total Glutathione Measurement

Intracellular glutathione levels were determined by Total Glutathione Assay Kit (Cat#S0052, Beyotime) according to manufacturer’s instructions. Briefly, 3 × 10^5^ cells were seeded in 6-well plates, and 1 × 10^6^ cells were collected and washed with DPBS 24 h after seeding; total glutathione levels were then analyzed according to the protocol.

### 2.8. PK Activity Assay

PK activity was determined by Pyruvate Kinase Activity Assay Kit (Cat#MAK072, Sigma-Aldrich, St. Louis, MO, USA) according to manufacturer’s instructions. In brief, 8 × 10^5^ cells were rapidly homogenized with 100 μL pyruvate kinase assay buffer. Centrifuge at 15,000× *g* for 10 min to remove insoluble materials. Then, 50 μL samples were added to 50 μL reaction mix in 96-well plates, and incubated at room temperature; we took the initial measurement 2–3 min after mixture, and then took sequential measurements at 5 min intervals.

### 2.9. Cell Extraction, SDS-PAGE, and Western Blotting

Cells were washed with PBS and lyzed with RIPA buffer (Cat#9806, Cell Signaling, Boston, MA, USA) containing PMSF (Cat#ST505, Beyotime). Protein concentration was determined by BCA assay (Cat#P0011, Beyotime); equal amounts of protein were run on 10% SDS-page gels and transferred onto a polyvinylidene difluoride membrane via wet transfer. Membranes were probed with primary antibodies and HRP-labeled secondary antibodies. Western-blotting analysis was performed using a standard protocol. Antibodies used are as follows: ISLR (Cat# HPA050811, Sigma-Aldrich, 1:1000 dilution), PKM2 (Cat#4053, Cell Signaling, 1:1000 dilution), TUBULIN (Cat#ab6046, Abcam, Cambridge, UK, 1:1000 dilution), FLAG (Cat#F1804, Sigma, 1:1000 dilution), goat anti-rabbit secondary antibody (Cat#A0277, Beyotime, 1:10000 dilution), and goat anti-mouse secondary antibody (Cat#A0286, Beyotime, 1:10000 dilution). Protein ladder (Cat#26619, Thermo Fisher) was also included in every experiment.

### 2.10. Disuccinimidyl Suberate (DSS) Cross Linking

Cells were collected and washed twice with ice-cold PBS (pH = 8.0); then, DSS crosslinker (Cat#HY-W019543, MedChemExpress) was added to a final concentration at 250 μM, incubated the mixture for 30 min at 37 °C, and then the reaction was quenched with 10 mM Tri-Hcl (pH = 7.5) for 15 min at room temperature. Cells were collected with centrifugation and lysed as above.

### 2.11. Immunoprecipitation Analysis

For immunoprecipitation assays, 3 × 10^6^ HEK293T cells were seeded into 10 cm plates. Then, 10 μg plasmids were transfected with Lipofectamine 2000, and, 48 h after transfection, cells were washed with DPBS twice, and then collected by IP lysis. Cell lysates were precleared with protein A/G beads (Cat# 88803, Thermo Fisher) for 1 h at 4 °C, and then incubated with antibody overnight at 4 °C with gentle rotation. Immunoprecipitates were washed 5 times with IP lysis after incubation and proteins separated by western blotting. Primary antibodies used in this assay were the same with antibodies used in western blotting.

### 2.12. Immunofluorescence and Crystal Violet Staining

For immunofluorescence staining, cells cultured on glass-bottom culture dishes were washed twice with PBS, and then fixed with 4% paraformaldehyde for 1 h. After fixation, cells were permeabilized for 10 min in 0.5% Triton X-100 (Cat# ST1723, Beyotime) in PBS, and then blocked in blocking buffer (Cat# P0260, Beyotime) for 1 h at room temperature. PKM2 antibody (Cat#4053, Cell Signaling) diluted in primary antibody dilution buffer (Cat#P0262, Beyotime) was used to incubate the cells overnight at 4 °C. Primary antibodies used in this assay are the same with antibodies used in western blotting. Subsequently, cells were washed 3 times with PBS, and then incubated with a secondary antibody Alexa Fluor 488 Polyclonal Antibody (Cat#A-11094, Invitrogen) for 1 h at room temperature. DAPI (Cat# C1005, Beyotime) was used to counterstain the nuclei after cells were washed 3 times with PBS. Fluorescent images were acquired using the Echo Revolve Hybrid optical microscope.

To determine the viability of cultured cells, crystal violet staining was performed as previously described [17]; cells cultured on 6-well plate were fixed as described above, washed twice with PBS, and then stained with crystal violet staining solution (Cat# C0121, Beyotime) for 10 min; and, after washing 3 times with PBS, bright-field images were acquired using the Echo Revolve Hybrid optical microscope.

### 2.13. Triple-Negative Breast Cancer Mouse Model

Triple-negative breast cancer mouse models were established by the following steps. First, 5 × 10^6^ cell/mouse 4T1 cells were injected into the right mammary fat pad of female Balb/c mice (6 weeks old, Vitalriver, Beijing, China) under anesthesia. All the mice were randomly divided into four groups 10 d after injection, followed with intraperitoneal injection of indicated reagents. Body weights and tumor volumes were recorded every 5 d; tumor volumes were measured using dial caliper. Then, 20 d post injection, all the mice were sacrificed under anesthesia and the tumor samples were collected for further analysis.

### 2.14. Statistical Analysis

All experiments included at least three biological replicates. Values are presented as the means ± SEM. A two-tailed Student’s test in Microsoft Excel 2016 was used to analyze statistical significance for experiments which have 2 groups, while one-way ANOVA and post hoc Tukey’s test (alpha = 0.05) in GraphPad Prism 10 were used for experiments which have 3 or more groups. Statistical significance was accepted at *p* < 0.05.

## 3. Results

### 3.1. ISLR Is A Redox Sensor

ISLR is known to be suppressed by hypoxia, which is a potential inducer of ROS [18,19]. To more comprehensively identify whether ISLR is a redox sensor of ROS, we treated four different cell lines, which including 4T1 (breast cancer cells), A549 (lung cancer cells), HEK293T (embryonic human kidney cells), and C3H10T1/2 (mesenchymal progenitor cells) cells with H_2_O_2_, a natural ROS produced by mitochondria. Expression analysis revealed that different cell lines differ in their tolerance to H_2_O_2_ treatment, and H_2_O_2_ treatment decreased the level of ISLR protein concentrations, without suppressing the *ISLR* mRNA expression (Appendix A), indicating that the modulation of ISLR concentrations occurs at the protein level, not the transcriptional level. The ectopic expression of ISLR in 293T cells and A549 cells further confirmed that H_2_O_2_ treatment suppressed the level of ISLR in a dose- and time-dependent manner (Figure 1A–C and Appendix A), and the decrease in ISLR level could be observed as early as 15 min (Figure 1A,B). To investigate whether the suppression of ROS on the ISLR concentration observed in ISLR-overexpressed cells can be extended to primary cells, we isolated mouse embryonic fibroblasts (MEFs); the detection of ISLR also confirmed that H_2_O_2_ treatment decreased the level of ISLR in a dose-dependent manner (Figure 1D). In addition, we tested the effect of another ROS inducer, diamide, on ISLR, and, as expected, the ISLR concentration was suppressed by diamide in a dose-dependent manner (Figure 1E). Boosting intracellular ROS production by disturbing the mitochondrial function with rotenone and antimycin A (inhibitors targeting electron transport chain complex I or III, respectively) or oligomycin (inhibitor targeting ATP synthase) treatment also decreased ISLR levels (Appendix A) in a time-dependent manner (Figure 1F). Moreover, the pretreatment with N-acetylcysteine (NAC), an ROS scavenger, completely reversed the suppression effect of ROS on ISLR (Figure 1G), demonstrating that the downregulation of ISLR following H_2_O_2_ treatment was induced by ROS. Together, these data demonstrated that ISLR is a redox sensor.

### 3.2. Cys^19^ in ISLR Is Required for Oxidative-Stress-Induced ISLR Degradation

To determine which fragment of the ISLR protein is responsible for sensing ROS, we constructed truncated mutants of ISLR (Appendix A) and found that all these truncated mutants were downregulated following H_2_O_2_ treatment, which indicate that the shortest fragment (1–168 aa) of ISLR is responsible for ROS sensing (Figure 1H). Since the oxidation of cysteine residues is known to be the main mechanism that mediates the response of protein to ROS [20,21,22,23], we analyzed cysteine residues within the 1–168 aa fragment of ISLR, and found there are four cysteine residues (Cys^19^, Cys^23^, Cys^25^, and Cys^36^) (Appendix A), and, by analyzing the residue conservation in *human*, *mice*, *rabbits*, and *rats*, we found the four residues are all highly conserved in these species (Appendix A). By mutating each cysteine residue to serine, we constructed four mutants of ISLR (Appendix A) and found that all the mutants were downregulated by H_2_O_2_ treatment except the C19S mutant (Figure 1I and Appendix A); this indicates that the Cys^19^ residue is required for the ROS sensing of ISLR.

Then, we examined the mechanism by which ROS downregulate the ISLR concentration; given that the autophagy–lysosomal pathway has been widely acknowledged to eliminate oxidative damage to macromolecules [24], we examined whether ISLR is degraded in the autophagy–lysosomal pathway. Two autophagy inhibitors, ammonium chloride and 3-methyladenine (3-MA), were applied to prevent the degradation of autophagosomes; the result showed that the downregulation of ISLR was reduced by a pretreatment with either inhibitor (Figure 1J). Together, these results demonstrate that oxidative-stress-induced ISLR degradation is in the autophagy–lysosome pathway.

### 3.3. ISLR Deletion Increases Antioxidative Capacity

Then we investigated whether ISLR can alter ROS levels in the presence of an ROS inducer. Remarkably, 4T1 and MEFs lacking *ISLR* exhibited decreased ROS levels compared to the control (Figure 2A,B and Appendix A), while ISLR-overexpression A549 cells had higher ROS levels (Figure 2C). In contrast to the ROS levels, ISLR deletion moderated a decreased cell viability in H_2_O_2_-treated cells (Figure 2D,E), while a decreased cell viability in response to H_2_O_2_ was found in ISLR-overexpressed cells (Figure 2F). Consistent with the previous findings, following H_2_O_2_ treatment, the expression of *Bax*, *Bak,* and *Caspase 8*, apoptotic genes known to be induced by ROS, were significantly increased in the control group, while *ISLR* deletion significantly moderated this increase (Figure 2G). *ISLR*-deleted 4T1 cells also exhibited decreased ROS levels when cells were exposed to oligomycin, or rotenone plus antimycin A (AA) (Figure 2H). Given that ROS production is known to be induced in a three-dimensional culture due to matrix detachment [25], we tested whether ISLR can moderate ROS levels under this condition, and found that *ISLR*-deleted 4T1 spheroids have a lower ROS level compared to the control (Figure 2I). In combination, these data indicate that ISLR is an important modulator of oxidative stress; *ISLR* deletion significantly increases the antioxidative capacity.

### 3.4. Regulation of Antioxidant Capacity by ISLR Deletion Is Not Mainly through GSH Production

As glutathione is a major antioxidant [26], the intracellular glutathione concentration was examined in *Islr*-deleted cells, and we found that *Islr*-deletion significantly increased glutathione levels in 4T1 and MEFs (Figure 3A,B). To determine whether these increased intracellular glutathione levels are the main reason that *Islr*-deleted cells have an increased antioxidant capacity, cells were pretreated with L-Buthionine-sulfoximine (L-BSO), a small molecular inhibitor of glutathione synthesis, and then treated with both L-BSO and H_2_O_2_; we found that, although L-BSO treatment significantly decreased the intracellular glutathione levels in both control and *Islr*-deleted cells (Figure 3C,D), the ROS levels in the deleted group were significantly lower than those in the control group (Figure 3E). To further confirm these results, Erastin, a ferroptosis inducer that can inhibit the exchange of intracellular glutamate and extracellular cystine to reduce the production of glutathione (Figure 3C,F), was used to induce ROS; *Islr* deletion significantly reduced the ROS levels in both H_2_O_2_- and Erastin-treated cells (Figure 3G). Together, these results indicate that, although *Islr* deletion increased total glutathione levels, the increased glutathione levels are not the principal reason that *Islr*deletion increased the antioxidant capacity.

### 3.5. ISLR Interacted with PKM2 and Downregulates PKM2 Tetramerization

To determine how ISLR regulates the antioxidative capacity, co-immunoprecipitation (Co-IP) was conducted to identify the enzymes that could potentially interfere with metabolism. Pyruvate kinase, a critical glycolytic enzyme that catalyzes the formation of pyruvate from phosphoenolpyruvate, was identified as a potential interacting protein. A co-IP assay confirmed that ISLR directly interacts with PKM2 (Figure 4A,B), and this interaction persisted under H_2_O_2_ treatment (Appendix A). qPCR results demonstrated that the mRNA levels of Pkm was not affected by *Islr* deletion (Appendix A). However, the ectopic expression of *ISLR* in HET293T cells significantly decreased the level of PKM2 (Appendix A). Immunofluorescence staining showed that the level of Pkm2 was upregulated in *Islr*-deleted 4T1 and C2C12 cells (Figure 4C and Appendix A), and downregulated in *ISLR*-overexpressed A549 cells (Figure 4D).

PKM2 exists as a monomer, dimer, and tetramer, and the tetramer is the principle state which maintains the active pyruvate kinase activity [27]. To assess whether ISLR modulates the status of PKM2 subunit association, crosslinking studies were performed. *ISLR* deletion was found to increase the Pkm2 level, as well as Pkm2 tetramerization (Figure 4E and Appendix A). To determine whether ISLR can affect PKM2 tetramerization when cells are exposed to oxidative stress, ISLR-overexpressed A549 cells were treated with H_2_O_2_ at different concentrations. A crosslinking assessment confirmed that PKM2 tetramerization was enhanced by H_2_O_2_ in a dose-dependent manner in the control group, but it was suppressed by ISLR overexpression (Figure 4F). To further confirm that ISLR can suppress PKM2 tetramerization, we constructed a *Pkm2*-3×Flag plasmid, and confirmed that the recombinant protein *Pkm2*-3×Flag has a larger stripe size than the endogenous PKM2 (Appendix A). The co-IP assay demonstrated that *Pkm2*-3×Flag can directly bind with endogenous PKM2, and ISLR overexpression reduced the amount of endogenous PKM2 which binds with *Pkm2*-3×Flag (Figure 4G and Appendix A), further demonstrating that ISLR suppresses PKM2 tetramerization. Consistent with this observed enhancement in PKM2 tetramerization, the PK activity was significantly increased in *ISLR*-deleted cells (Figure 4H). In combination, these results demonstrate that ISLR suppresses PK activity by directly interfering with PKM2 tetramerization.

### 3.6. PKM2 Is Critical to ISLR-Mediated Antioxidative Effect

As pyruvate is the product of pyruvate kinase, and pyruvate supplementation is widely known to prevent oxidative stress [28,29,30,31], to eliminate the effect of the pyruvate supplement, 4T1 cells were cultured in RPMI-1640 culture medium without pyruvate. To examine whether PKM2 is essential for the *ISLR*-deletion-moderated antioxidative effect, cells were then treated with shikonin (a specific PKM2 inhibitor which can reduce the production of pyruvate from phosphoenolpyruvic acid). Shikonin treatment was found to significantly increase the intracellular ROS levels in both control and *Islr*-deleted cells, and the enhanced antioxidative capacity of *Islr*-deleted cells were completely negated by shikonin treatment. Moreover, a pyruvate addition was found to fully prevent the increased ROS levels induced by H_2_O_2_ and shikonin (Figure 5A). Similar results were also observed in cells treated with another PKM2 inhibitor, PKM2-IN-1 (Figure 5B). These results indicate that *Islr* deletion enhances the antioxidative capacity via altering the PKM2 activity. To further confirm this result, live cell numbers were determined, and the result showed that, in contrast to ROS levels, both shikonin and PKM2-IN-1 treatment significantly reduced live cell numbers, and negated the effect of *Islr* deletion (Figure 5C). In addition, enhancing PKM2 activity by pretreating cells with the PKM2 activator DASA-58 increased the antioxidative capacity (Figure 5D).

Given that PKM2 is known to affect the antioxidative capacity via both enzymatic and non-enzymatic mechanisms, to confirm whether glycolysis is critical to the *Islr*-deletion-enhanced antioxidative capacity, 4T1 cells were pretreated with 2-deoxy-d-glucose (2-DG), a glucose analogue which inhibits glycolysis, followed by a combined treatment of both 2DG and H_2_O_2_. The 2DG treatment was found to significantly increase intracellular ROS level in both control and *Islr*-deleted cells, and completely negated the increased antioxidative capacity in *Islr*-deleted cells (Figure 5E). In contrast to the ROS levels, a cell viability assay and crystal violet staining also confirmed that 2DG significantly decreased the cell viability and cell survival in both control and *Islr*-deleted cells (Figure 5F,G), and a pyruvate supplement can fully mitigate the decreased cell survival which was induced by H_2_O_2_ treatment (Figure 5G). Similar results were also observed in cells which were treated with Bay-876, a glucose transporter inhibitor which can inhibit both glycolysis and the pentose phosphate pathway (Figure 5H,I). Taken together, these results demonstrate that *Islr*-deleted cells achieve an increased antioxidative capacity via PKM2-mediated glycolysis; PKM2 activity is critical to the ISLR-mediated antioxidative effect.

### 3.7. PKM2 Inhibition Sensitized Triple-Negative Breast Cancer to Cisplatin

Cisplatin is a well-known chemotherapeutic drug which is widely used for the treatment of triple-negative breast cancer. As previous reports have demonstrated that cisplatin can induce oxidative stress [31], to determine whether ISLR can respond to cisplatin treatment in triple-negative breast cancer in vivo, we transplanted 4T1 cells into Balb/c mice and found that cisplatin treatment significantly reduced ISLR protein in the tumor tissue (Figure 6A), demonstrating that ISLR can respond to cisplatin treatment in triple-negative breast cancer in vivo.

As *Islr* deletion can enhance the antioxidative capacity by increasing the PKM2 activity, we investigated whether breaking the link between ISLR and PKM2 can increase the sensitivity of triple-negative breast cancer cells to cisplatin by elevating ROS levels; therefore, 4T1 cells were treated with shikonin, cisplatin, or both reagents. Compared to untreated cells, both shikonin or cisplatin alone was found to trigger the increase in ROS levels and the decrease in cell numbers. However, the combined shikonin and cisplatin treatment induced considerably higher ROS levels and lower cell numbers (Figure 6B,D and Appendix A). Similar results were also observed in cells treated with another PKM2 inhibitor PKM2-IN-1 (Figure 6C,E and Appendix A). To test whether PKM2 inhibition sensitizes triple-negative breast cancer cells to cisplatin in vivo, mice with breast cancer were injected separately with shikonin, cisplatin, or both reagents (Figure 6F). Cisplatin treatment significantly reduced the tumor volume and tumor weight, while shikonin treatment only had a minor effect. The combined treatment also significantly reduced the tumor volume and tumor weight (Figure 6G–I). The latter treatment did not reduce body weight compared to the cisplatin-treated group (Figure 6J). These findings demonstrate that the in vivo shikonin treatment can increase the vulnerability of triple-negative breast cancer to cisplatin, and PKM2 might be a potential target to sensitize triple-negative breast cancer cells during chemotherapy.

## 4. Discussion

In this paper, we provide the first evidence that, upon oxidative stress, ISLR serves as an ROS sensor that perceives the change in ROS levels through Cys^19^ in its N-terminal LRRNT domain; then, it rapidly degraded in the autophagy–lysosomal pathway. Since Cys^19^ is highly conserved among vertebrates, the decreased ISLR protein may serve as a hallmark of oxidative stress among species. Moreover, we found that ISLR is a critical modulator of ROS defenses; under normal conditions, ISLR suppresses the tetramerization of PKM2, therefore maintaining the PKM2 activity at a low level. However, when cells are exposed to oxidative stress, the existence of ROS rapidly degrades ISLR and, therefore, enhances the tetramerization of PKM2, promotes PKM2-mediated glycolysis, and, eventually, enhances the intracellular antioxidant capacity. These results reveal a novel mechanism by which ISLR regulates the antioxidant capacity (Figure 7).

An interesting finding of this study is that ISLR functions as an ROS sensor to perceive the changes in ROS levels: although we proved that Cys^19^ in the N-terminal LRRNT domain of ISRL is critical for ISLR to sense ROS levels, it remains unclear under oxidative stress whether this residue is oxidized or not; it will be worthwhile to investigate how this reside is critical to ROS sensitivity. Moreover, ISLR degradation could be observed as early as 15 min; although, by treating cells with autophagy inhibitors (ammonium chloride and 3-methyladenine), we found that the degradation of ISLR is greatly rescued, it still has the possibility that ubiquitin-dependent proteolysis or other degradation pathways might also participate in the degradation of ISLR.

PKM2 is a rate-limiting glycolytic enzyme that catalyzes the conversion of phosphoenolpyruvate to pyruvate, and it is widely expressed in normal tissues like embryonic tissue, intestine, spleen, ovaries, and so on [32,33]. Moreover, PKM2 is also known to be upregulated in most human cancers [34]; however, the function of PKM2 in the antioxidative capacity remains unresolved. Anastasiou et al. demonstrated that the inhibition of PKM2 diverts the glucose flux into the pentose phosphate pathway, and thereby generates more NADPH to eliminate ROS [15]. However, Wei et al. showed that the enhanced PKM2 dimerization facilitates glutathione production by binding to Nrf2 [35]. Mao et al. reported that shikonin treatment significantly increased ROS levels [36]. In contrast, our results showed that the PKM2 inhibitor moderated the cellular antioxidant capacity. We presume that the different culture medium might be responsible for this difference in antioxidant capacity. Although it has been demonstrated that a pyruvate supplement can provide a protective effect in cultured cells [28,30], the secretion of pyruvate is reported to be an antioxidant defense in mammalian cells [37]. However, intracellular pyruvate is regarded as a weak antioxidant [38]. In our study, *Islr-*deleted cells achieved an enhanced antioxidative capacity by promoting PKM2 activity, whereas pyruvate is known to directly eliminate ROS though its conversion to acetate [39]. We propose that intracellular pyruvate can function as a critical antioxidant, and the enhanced antioxidant capacity of PKM2 activation can be achieved by enhancing pyruvate production. Based on this proposition, sufficient extracellular pyruvate compensates for the reduced antioxidant capacity due to the pyruvate-deficiency-mediated PKM2 inhibition. In our study, all the cells used for ROS detection were cultured in pyruvate-free or -limited culture medium; this might enable us to fully determine the function of PKM2 in antioxidative activity. The differences in the cell culture medium might be the reason we did not observe that the inhibition of PKM2 diverted the glucose flux into the pentose phosphate pathway, and generated more NADPH to eliminate ROS [15].

It is noteworthy that, although we proved that ISLR can regulate the pyruvate kinase activity by suppressing the tetramerization of PKM2, the reason why the total PKM2 level changed remains unclear. There might be other mechanisms for ISLR to regulate the expression of PKM2. Moreover, our results proved that *Islr* deletion enhanced the production of total glutathione; although GSH did not prove to be a critical factor for the enhanced antioxidant capacity in *Islr-*deleted cells, the mechanism by which ISLR regulates the total glutathione level remains unknown. As PKM2 is known to induce glutathione synthesis via Nrf2 transactivation [35], ISLR-mediated PKM2 polymeric formation might be the reason that *Islr-*deleted cells have higher glutathione levels.

It is also well-known that most chemotherapies can induce ROS production resulting in cell death. However, cancer cells have developed mechanisms to resist oxidative stress. According to these findings, we propose that ISLR is involved in the enhancement of the cellular antioxidant capacity when cancer cells are treated with cisplatin, and, as expected, blocking PKM2 activity is sufficient to sensitize triple-negative breast cancer to cisplatin both in vitro and in vivo. Despite the increasingly appreciated role of shikonin in sensitizing tumor cells to cisplatin in cancers like non-small cell lung cancer [40], and bladder cancer [41], the combined administration of both shikonin and cisplatin was not evaluated in triple-negative breast cancer yet; our finding demonstrates the potential clinical utility of targeting PKM2 in triple-negative breast cancer therapy.

## 5. Conclusions

ISLR is a redox sensor to perceive the fluctuation of intracellular oxidative stress, and regulate antioxidant capacity by suppressing PKM2 activity. Combined treatment of PKM2 inhibitor and chemotherapy is a potential therapy for triple-negative breast cancer.

## Figures and Tables

**Figure 1 cells-13-00838-f001:**
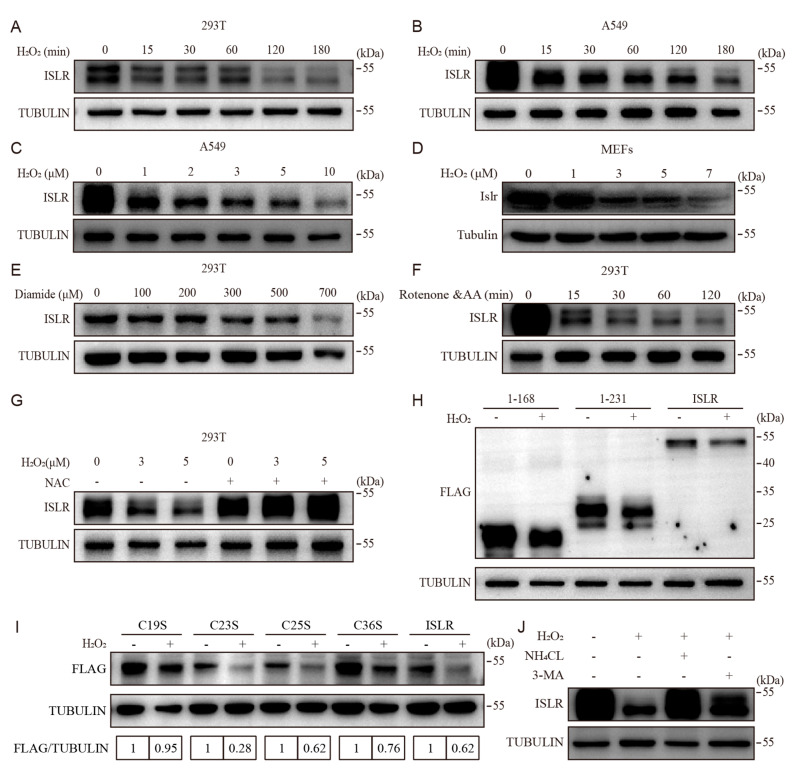
ISLR is a redox sensor. (**A**,**B**) Western blot analysis of ISLR in HEK293 cells (**A**) and A549 cells (**B**). HEK293 cells which transiently expressed *ISLR* were treated with H_2_O_2_ at 7 μM for indicated times; A549 cells which stably expressed *ISLR* were treated with H_2_O_2_ at 2 μM for indicated times. (**C**,**D**) Western blot analysis of ISLR in A549 cells and mouse embryonic fibroblasts (MEFs) which were treated with H_2_O_2_ at indicated dose for 3 h, with A549 cells (**C**) stably expressing *ISLR*, while MEFs (**D**) were isolated from wild-type mice. (**E**) Western blot analysis of ISLR in HEK293 cells which transiently expressed *ISLR* and were treated with diamide at indicated dose for 3 h. (**F**) Western blot analysis of ISLR in HEK293 cells which transiently expressed *ISLR* and were treated with 10 μM rotenone and antimycin A for the indicated time. (**G**) Western blot analysis of ISLR in HEK293 cells which transiently expressed *ISLR*, and were pretreated with 10 mM NAC for 3 h, followed by combined treatment with 10 mM NAC and 7 μM H_2_O_2_ at indicated dose for 3 h. (**H**) Western blot analysis of FLAG in HEK293T cells which expressed *Islr*-3×Flag or its mutant derivatives (1-168, 1-231) and were treated with H_2_O_2_ for 3 h. (**I**) Western blot analysis of FLAG in HEK293T cells which expressed *Islr*-3×Flag or its mutant derivatives (C19S, C23S, C25S, C36S) and were treated with H_2_O_2_ for 3 h. (**J**) Western blot analysis of ISLR in A549 cells which stably expressed *ISLR*, and were pretreated with 10 mM NH4CL or 3-MA for 3 h, followed by treatment with 2 μM H_2_O_2_ for 3 h**.**

**Figure 2 cells-13-00838-f002:**
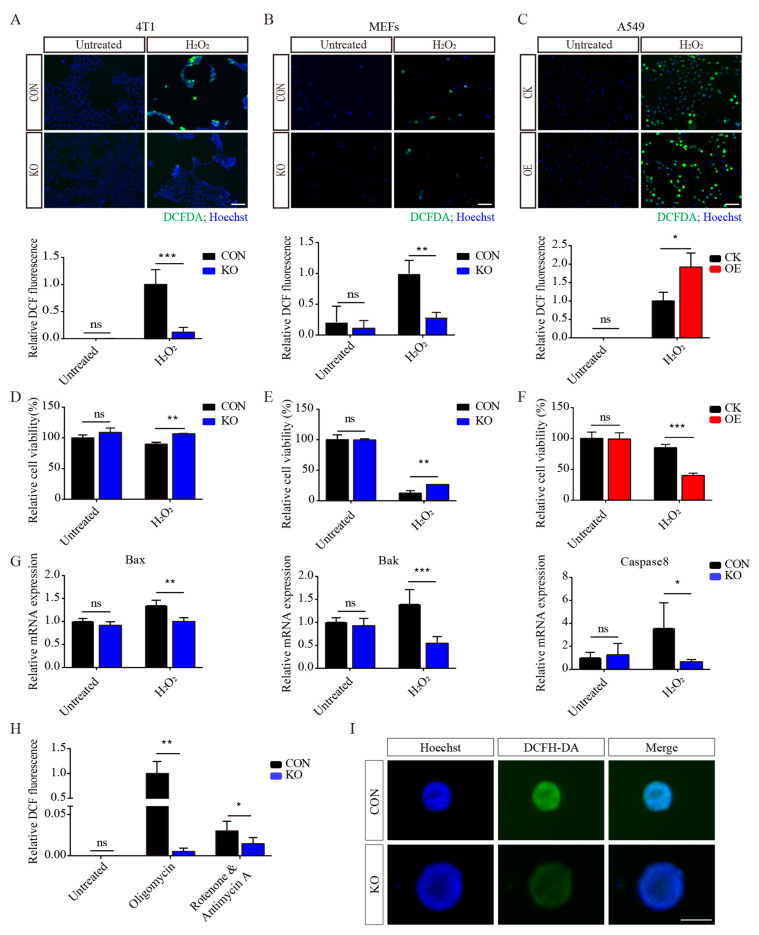
ISLR decreases antioxidative capacity in response to oxidative stress. (**A**–**C**) Representative pictures (Upper) and quantification (Lower) showing changes in ROS levels of 4T1 (**A**), MEFs (**B**), and A549 (**C**) cells. The 4T1 and A549 cells were treated with 2 μM H_2_O_2_ for 6 h, and MEFs were treated with 1 μM H_2_O_2_ for 6 h. CK group included cells which were stably transfected with vector plasmid; OE group included cells which were stably transfected with *ISLR* overexpression plasmid. (**D**–**F**) Relative cell viability of 4T1 (**D**), MEFs (**E**), and A549 (**F**) cells; cells were treated as described in A, B, and C. (**G**) The 4T1 cells were treated with 2 μM H_2_O_2_ for 6 h; cells were analyzed with quantitative reverse-transcription polymerase chain reaction (qRT -PCR) for indicated genes. (**H**) Relative ROS level measurement of 4T1 cells treated with 10 μM oligomycin or rotenone & antimycin A for 6 h. (**I**) The 4T1 cells were cultured using 3D culture plate; ROS levels were determined by fluorescence of oxidized dichlorofluorescein diacetate (DCFDA) and hoechst**.** Scale bar = 100 μm. * *p* < 0.05; ** *p* < 0.01; *** *p* < 0.001; ns, not significant.

**Figure 3 cells-13-00838-f003:**
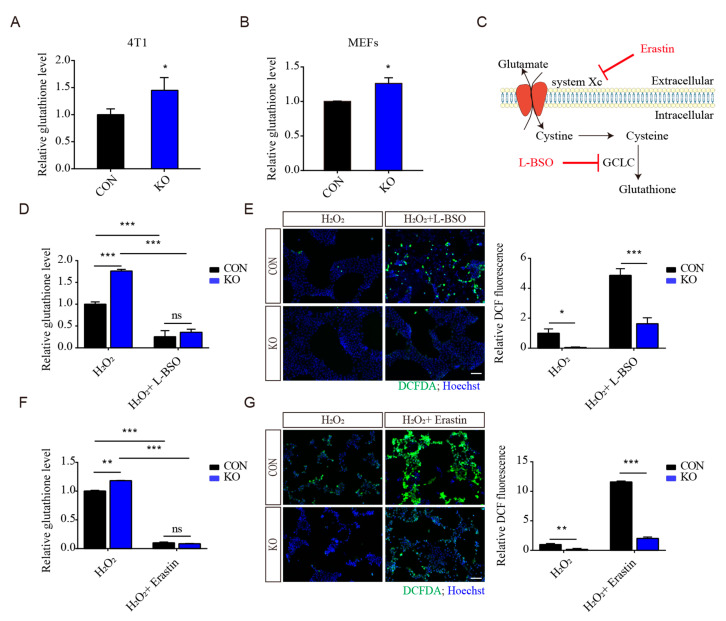
The regulation of antioxidant capacity by ISLR is not through GSH production. (**A**,**B**) Relative total glutathione levels were determined for control and *Islr*-knockout 4T1 cells (**A**) and MEFs (**B**). (**C**) Schematic diagram of glutathione production and the functions of L-BSO and Erastin. (**D**,**E**) Relative total glutathione levels (**D**) and ROS levels (**E**) were determined for control and *Islr*-knockout 4T1 cells which were treated with 1 mM L-BSO for 3 h, followed by combined treatment of both 1 mM L-BSO and 2 μM H_2_O_2_. (**F**,**G**) Relative total glutathione levels (**F**) and ROS levels (**G**) were determined for control and *Islr*-knockout 4T1 cells which were treated with 10 μM Erastin for 3 h, followed by combined treatment of both 10 μM Erastin and 2 μM H_2_O_2_. Scale bar = 100 μm. * *p* < 0.05; ** *p* < 0.01; *** *p* < 0.001; ns, not significant.

**Figure 4 cells-13-00838-f004:**
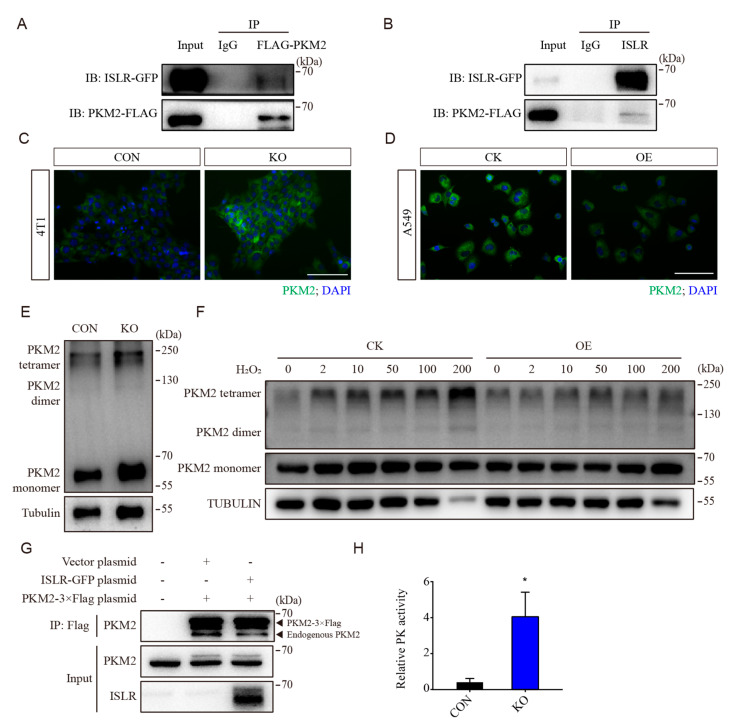
ISLR downregulates PKM activity by suppressing PKM2 tetramerization. (**A**,**B**) Reciprocal co-immunoprecipitation analysis between *ISLR*-GFP and *PKM2*-Flag in HEK293T cells cultured for 2 d; Flag antibody (**A**) or ISLR antibody was used as the bait. (**C**,**D**) PKM2 level in 4T1 cells (**C**) and A549 cells (**D**) were analyzed using immunofluorescence staining (IF). (**E**) Western blot analysis of PKM2 dimer and tetramer in DSS crosslinked 4T1 cells. (**F**) Western blot analysis of PKM2 dimer and tetramer in DSS crosslinked A549 cells which stably expressed *ISLR* and were treated with H_2_O_2_ at indicated dose for 30 min. (**G**) Reciprocal co-immunoprecipitation analysis between endogenous *Pkm2* and *Pkm2*-3×Flag in HEK293T cells with or without ISLR overexpression; cells were cultured for 2 d, followed with treatment of 7 μM H_2_O_2_ treatment for 3h; FLAG antibody was used as the bait. (**H**) Relative pyruvate kinase activity was analyzed in 4T1 cells. Scale bar = 100 μm. Data are means ± SEM (n = 4). * *p* < 0.05.

**Figure 5 cells-13-00838-f005:**
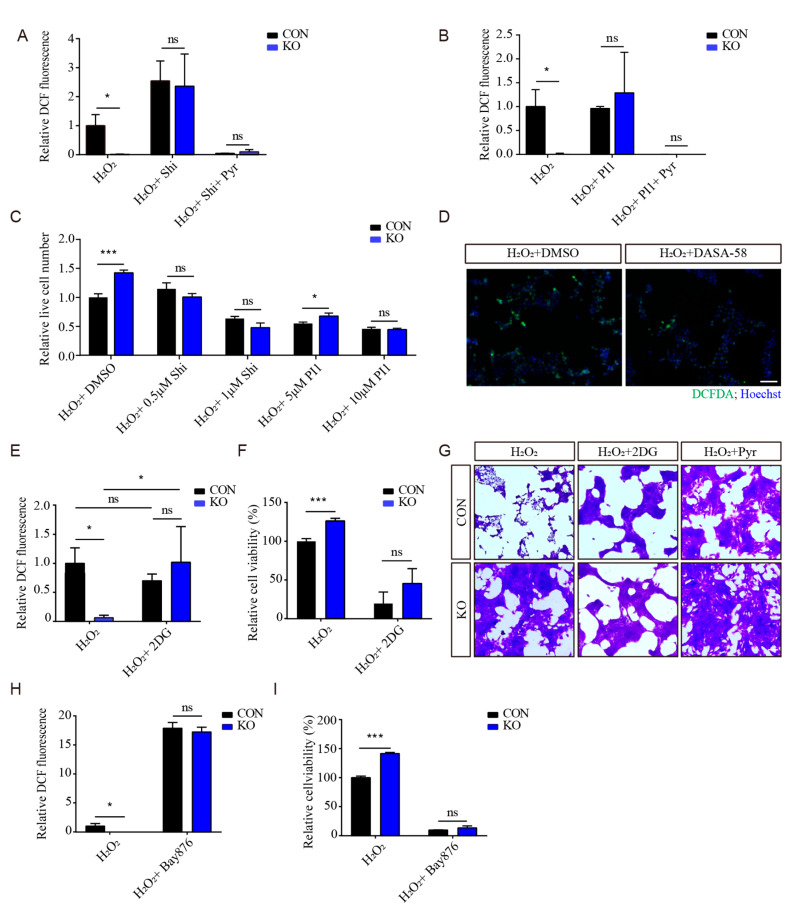
PKM2 is critical to ISLR-mediated antioxidative capacity. (**A**) Relative ROS level measurement of 4T1 cells which were pretreated with 1 μM shikonin for 3 h, followed by combined treatment with 1 μM shikonin and 2 μM H_2_O_2_ with/without 1mM pyruvate for 6 h. (**B**) Relative ROS level measurement of 4T1 cells which were pretreated with 10 μM PKM2-IN-1 for 3 h, followed by combined treatment with 10 μM PKM2-IN-1 and 2 μM H_2_O_2_ for 6 h. (**C**) Relative live cell number measurement of 4T1 cells which were pretreated with shikonin or PKM2-IN-1 at the indicated dose for 3 h, followed by combined treatment with shikonin or PKM2-IN-1 and 2 μM H_2_O_2_ for 6 h. (**D**) The 4T1 cells were pretreated with 25 mM DASA-58 for 3 h, followed by combined treatment with 25 mM DASA-58 and 2 μM H_2_O_2_ for 6 h; ROS levels were determined by fluorescence of oxidized DCFDA and hoechst. (**E**,**F**) Relative ROS levels (**E**) and cell viability (**F**) measurement of 4T1 cells which were pretreated with 5 mM 2DG for 3 h, followed by combined treatment with 5 mM 2DG and 2 μM H_2_O_2_ for 6 h. (**G**) Crystal violet staining of 4T1 cells which were pretreated with 5 mM 2DG for 3 h, followed by combined treatment with 5 mM 2DG and 2 μM H_2_O_2_ for 6 h. (**H**,**I**) Relative ROS levels measurement (**H**) and cell viability (**I**) of 4T1 cells which were pretreated with 1 μM Bay876 for 3 h, followed by combined treatment with 1 μM Bay876 and 2 μM H_2_O_2_ for 6 h. Shi = Shikonin; Pyr = Pyruvate; PI1 = PM2-IN-1. Scale bar = 100 μm. Data are means ± SEM (n = 3); * *p* < 0.05; *** *p* < 0.001; ns, not significant.

**Figure 6 cells-13-00838-f006:**
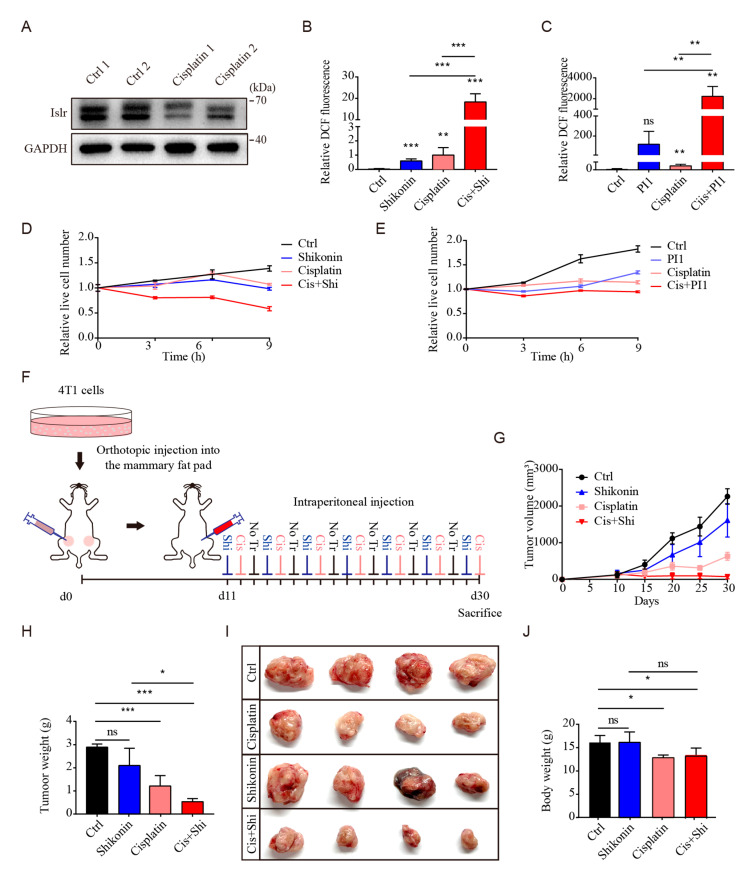
PKM2 inhibition sensitizes tumor cells to cisplatin. (**A**) ISLR level examined in tumor tissues from Balb/c mice; Balb/c mice were transplanted with 4T1 cells, and, 20 days later, 3 mg/kg cisplatin were intraperitoneal injected into the mice for 20 days. (**B**) Relative ROS levels measurement of 4T1 cells which were treated with 1 μM shikonin/1 μg/mL cisplatin or both reagents for 9 h. (**C**) Relative ROS levels measurement of 4T1 cells which were treated with 10 μM PKM2-IN-1/1 μg/mL cisplatin or both reagents for 9 h. (**D**,**E**) Relative live cell number of 4T1 cells which were treated with 1 μM shikonin/10 μM PKM2-IN-1, 1 μg/mL cisplatin, or both reagents for 9 h. (**F**) Experimental procedure presenting 4T1 cells transplantation into Balb/c mice and the dosing regimen. (**G**) Growth of tumors generated from 4T1 bearing Balb/c mice that were treated without (Vehicle) or with shikonin/cisplatin/both shikonin and cisplatin as indicated in (**F**). (**H**,**I**) Tumor weight (**H**) and tumor size (**I**) from tumor-bearing mice which were treated with different reagent. (**J**) Body weight of tumor bearing mice which were treated with different reagent. Data are means ± SEM (n = 3). Shi = Shikonin; Cis = Cisplatin; PI1 = PKM2-IN-1. * *p* < 0.05; ** *p* < 0.01; *** *p* < 0.001; ns, not significant.

**Figure 7 cells-13-00838-f007:**
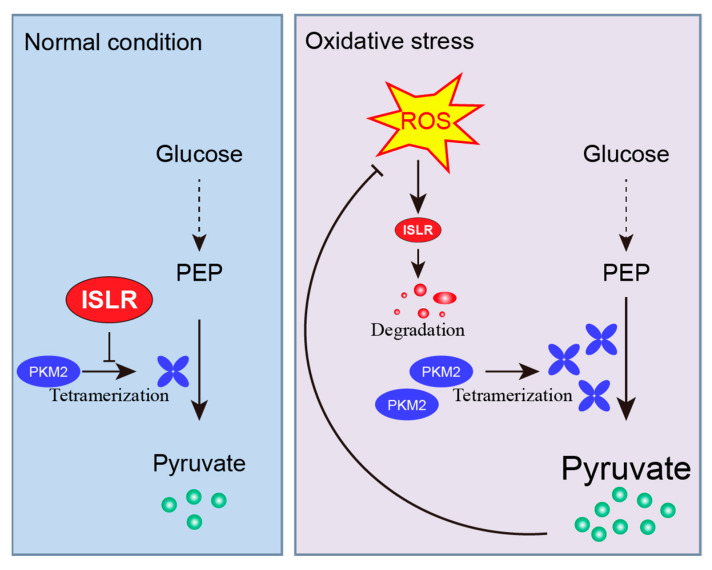
Mechanism of ISLR-mediated antioxidative capacity. A schematic model of oxidative-stress–induced ISLR degradation and ISLR-mediated redox balance. Under normal condition, ISLR suppresses the tetramerization of PKM2, and, therefore, decreases the pyruvate kinase activity. However, under oxidative stress, ISLR perceives the changes in ROS levels and rapidly degraded in the autophagy–lysosomal pathway. Without the suppression of ISLR, pyruvate kinase activity is increased by enhanced PKM2 tetramerization; elevated pyruvate level then improves intracellular antioxidative capacity.

## Data Availability

The raw data supporting the conclusions of this article will be made available by the authors upon request.

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
