# Peer review of "Immunoglobulin Superfamily Containing Leucine-Rich Repeat (ISLR) Serves as a Redox Sensor That Modulates Antioxidant Capacity by Suppressing Pyruvate Kinase Isozyme M2 Activity"

_cells, 2024, doi:10.3390/cells13100838_

Round 1

Reviewer 1 Report

Comments and Suggestions for Authors

COMMENTS FOR THE AUTHORS:

The authors of this article have demonstrated
the important role of the immunoglobulin superfamily containing leucine rich repeat (Islr) in regulating oxidative stress in various cancer cells and mice. Based on their results, the authors suggested that Islr is a potential therapeutic candidate. The authors showed expert skills in conducting the experiments and the interpretation of the results seems to be reasonable to this reviewer. This original article seems to contain very important and novel mechanisms about the function of Islr and its interacting partner pyruvate kinase isozyme M2 (PKM2) in regulating the levels of oxidative stress and a potential usage of targeting PKM2 in cancer therapy.

However, a few points listed below need to be double-checked to improve the current manuscript.

General comments:

1)    First of all, the apparent molecular size of the immunoblot bands should have been described on the right side of each gel throughout the manuscript.

2)    The authors should be careful not to use non-widely accepted abbreviated words (e.g., ISLR and PKM2) in the title and Abstract without full definitions. The full-names of these abbreviated words (MEFs, DSS, MCE, etc.) need to be spelled out when they appeared first in the text.  

3)    The authors should describe the functional roles of ISLR and PKM2 in tissue distribution and their roles under normal and disease states, including cancer, for the general readers.

4)    This reviewer is surprised to see the fast rapid degradation of ISLR via autophagy-lysosomal degradation. Thus, it would be desirable for the authors to describe the properties of autophagy-lysosomal degradation compared to ubiquitin-dependent proteolysis.

5)    Figure 2B: The intensity of DCFDA fluorescence in Fig 2B is a lot weaker than that of Fig 2A or Fig 2C. Yet, the relative ROS levels shown the y-axis of the three figures (bottom panels) are inconsistent with the top images. In addition, the y-axis in the top panels showed CK or OE but the bottom panels revealed Con or KO. These inconsistent ROS levels and labels should be corrected. 

6)    Figure 2E: The MEF cells look very sensitive to hydrogen peroxide-mediated cell death compared to those of the 4T1 cells. Please discuss any reasons for the large different sensitivity.

7)    Figure 2G: In general, various caspases, known as cell damage/apoptosis markers, are activated by proteolytic cleavage. Thus, the mRNA levels of caspase 8 may not be physiologically relevant while cleaved caspase-8 may be more important.

Minor comments:

1.    Line 34: it is known that ROS oxidizes KEAP1 to release NRF2 from KEAP1 binding, leading to the activation of NRF2. Is it true that oxidation of KEAP1 to inhibit the degradation of NRF2? This sentence needs to be double-checked.

2.    Lines 47-48: Please confirm whether Islr is induced by Nrf2 depletion [12], since this reviewer could not find the word Islr in this reference.

3.    Lines 64: 4T1 cell was provided – should have been: 4T1 cell line was ---. 

4.    Lines 89, 92, 100, 118, 119, 123, 129, 134, 151, 173, etc.: Please replace the regular number (3 x 104 cells) with superscripts (3 x 104 cells).

5.    Line 101: Please describe what the concentration of H2O2 was.

6.    Line 131: [U-13C] should have been [U-13C] superscript.

7.    Lines 157 & 167 & Fig 5G: It would be better for the authors to describe the reason why cresyl violet dye was used in cell staining, since it is usually used for staining of neuronal cells. The Fig. 5G images are unclear and may need to be replaced to improve the image quality.

8.    Line 382 & incomplete legends: Please complete the legends for Figs. 5H & I.

9.    Line 395: Please describe the full words of PPP and its function for the readers.

Comments on the Quality of English Language

It is fine.

Reviewer 2 Report

Comments and Suggestions for Authors

Please see below reviewer comments for the following manuscript, "ISLR serves as a redox sensor that modulates antioxidant capacity by suppressing PKM2 activity".

*Please provide the rationale for including both the cell viability assay and live number cell measurements.  Also, were these tests MTT or BrdU based?  Please confirm.

*There's critical information of multiple items in the methods missing, such as vendors, cat #'s, location, etc.  To enhance rigor and reproducibility, please include this omitted information.

*There is crucial information not present in the Stats section.  Please provide software and other tests that may have been utilized, such as if non-parametric tests being performed.  Some tests in the results also seem to include >2 groups, which t-tests won't suffice, so please add what stats were conducted in these instances.

*Please include the MW markers for all immunoblots.  Also, please quantify all immunoblots, so that statistical analyses may be performed.  Without this, the results are only descriptive and semi-quantitative at best, which weakens the results and makes data interpretation challenging.

*Some of the immunofluorescent staining is very poor, as you cannot conclude anything of value to such inferior images.  Please correct, so that data can be objectively assessed.  Also, the methods are lacking regarding cellular imaging, so please include more detailed methodology.

*For the working model in Figure 7, please provide more information within the figure legend to guide the reader through this figure better.

Comments on the Quality of English Language

English language is adequate/sufficient for an initial/first manuscript submission.

Reviewer 3 Report

Comments and Suggestions for Authors

In the present manuscript, the authors investigate the cellular accumulation of ISLR protein during stress caused by reactive oxygen species (ROS). Levels of ISLR are found to be decreased upon (ROS) exposure. One of the cysteine residues of ISLR appears to be sensitive to ROS and signals the degradation by autophagy. Here, some details are missing. There is no evidence for the oxidation of the indicated cysteine. It is also not known if ISLR is degraded through bulk autophagy with aggregated proteins/organelles or if it is selective chaperone-assisted autophagy? Moreover, ISLR is a secreted protein – to what extent does secretion affect its cellular levels?

ISLR is found to have a negative impact on cell resistance to ROS. To explain this phenotype, the authors indicate that a glycolytic enzyme, PKM2, is a molecular partner of ISLR. However, this evidence is not convincing and needs to be strengthened. Authors claim that ISLR negatively affects PKM2 levels and multimerization. Yet these effects appear mild in comparison to the reported impact on pyruvate kinase activity. A decrease in pyruvate kinase is proposed to be responsible for ROS sensitivity of ISLR-expressing cells.

Finally, the authors show that PKM2 inhibitor Shikonin increases cell sensitivity to cytostatic drug Cisplatin. This last observation is not strongly linked with the earlier investigation and confirms already published observations: doi: 10.1080/21655979.2022.2086378 or DOI: 10.1016/j.phymed.2023.154701 .

I find the investigation interesting, but some of the claims are not sufficiently evidenced, and some of the methods are questionable or unclear. Thus, I can not recommend the publication of the manuscript in its current form.

Figures accompanying the manuscript suffer from overuse of abbreviations, which are unnecessary as the space is not limited. Also, more experimental details should be included in the figures to help their interpretation. It is also unclear when native and ectopically expressed or fusion proteins are investigated. The legends also do not provide sufficient information on the experiments. Finally, some language editing is required.  

Below I list some specific points:

 The authors mention using mass spectrometry to identify ISLR’s molecular partners. However, these results are missing from the study, thus preventing interpretation by the reader.

Expression of ISLR is highly tissue-specific, with much higher mRNA levels in the endometrium and ovary. Are these tissues more ROS-sensitive?

Line 190 “significantly” suggests a result of statistical analysis. However, western blot results were not statistically analyzed. Please rephrase.

Lines 202-204) Antimycin A acts on complex III and not IV; Oligomycin inhibits ATP synthase and not complex III. Also, correct “retenone” -> rotenone.

Figure 1I supports the involvement of the cysteine residue 19 in the ROS-regulated proteolysis of ISLR. However, densitometry data is presented only for a single western blot result. Alco C19S mutant appears to be affected by H2O2 treatment. To support their claim, authors should provide averaged results from experimental repeats and their statistical analysis. 

Fig S1 A-D information on what is on the graphs is missing (what relative to what?)

Figure 2 Hoechst dye was used to stain DNA. Why is the Hoechst signal hardly visible in B and C and well visible in A? Why does the Hoechst signal have a doted distribution in C, which is different from A, and does not resemble nuclei? Why does the Hoechst signal differ between +/- ROS in A?

Figure 2 B) Micrographs mention OE/CK, while the graph mentions CON/KO. Which is correct?

Figure 2 uses many abbreviations that are not fully explained in the legend, some of which seem unnecessary as there is sufficient space on the figures.

Figure 2 Panels should indicate the cell line used.

Figure 2 What is represented on the graphs, and what are the whiskers? What was N?

Figure 2 A, B, and C graphs should be labeled relative DCF fluorescence and not relative ROS levels.

Figure 2 D,E,F relative luminescence of what?

Figure 2 H Is it also microscopy-based as in A,B,C?

Figure 2 Please explain the scale bars in the legend

Figure S3 A Expression of what? KO of what?

Figure 4 A,B As western blot signals for interacting proteins are weak, co-IP experiments require additional controls to prove their specificity. A) should include a sample without PKM2-FLAG B) without ISLR-GFP. Why was ISLR-GFP used if both IP and IB use anti-ISLR antibodies?

Figure 4 suffers from insufficient labeling. Panels should include information on the cells expressing the fusion proteins used. IB fragments should indicate both the antibody used and the full name of the detected protein (for fusion proteins). Also, please indicate kDa marker on large blot fragments.

Line 312 and Figure S3 B) I don’t see a significant decrease in PKM2 levels. Please provide densitometry with proper statistics. Also, could transfection with ISLR overexpression plasmid affect the efficiency of subsequent transfection with PKM2-expressing plasmid?

Fig 4E Why is the DAPI signal lower in “OE” cells?

Figure 4F) Authors mention that tetramer is the major form of cellular PKM2 (line 330), but the result indicates that monomer is the dominant form by at least tenfold. How could this be interpreted? Was crosslinking inefficient? How could this affect the interpretation of other results?

Figure 4H) Observed change is very small. Statistical analysis is required to test for significance.

Figure 4I How could minimal changes in PKM2 levels/tetramerization explain ~20-fold increase in activity? Was the assay specific to this enzyme?

Figure 5 A,B,G Pyruvate is a known quencher of H2O2 in a PKM2-independent manner (https://doi.org/10.1016/S0891-5849(97)00113-5). Thus, its addition to the assay might be confusing. Also, how does it affect earlier measurements of ROS? Was pyruvate present in the media?

Fig S1 legend – please expand AA.

Comments on the Quality of English Language

requires moderate editing for grammatical corrections. It is relatively easy to understand.

Round 2

Reviewer 2 Report

Comments and Suggestions for Authors

My only other concern is that the author needs to also add what post-hoc test(s) were performed following each one-way ANOVA test

Comments on the Quality of English Language

English language editing is needed.

Author Response

Comment: My only other concern is that the author needs to also add what post-hoc test(s) were performed following each one-way ANOVA test.

Response: Thank you for your comment, we have added " while one -way ANOVA and post hoc Tukey's test (alpha = 0.05) in GraphPad Prism was used for experiments which have 3 or more groups. " in the revised manuscript. -line 196-198.

Comments on the Quality of English:English language editing is needed.

Response: Thank you for pointing this out, we have checked the manuscript by a colleague fluent in English writing, and made some corrections.

Reviewer 3 Report

Comments and Suggestions for Authors

The authors addressed most of my concerns in the revised manuscript. Data presentation now benefits from the detailed description in the figure legends.

Due to many corrections, I strongly recommend the authors double-check if the figure panels match their descriptions in the legends and the results. For example, in the response to my question 2, the authors mention adding quantification to Fig S3C. This is also mentioned in the S3 legend (S3D) but does not appear on the figure itself. In the response, the authors also mention quantification for Figure 4C, but this panel was actually removed from the updated Figure 4. Such inconsistencies made reviewing more difficult.

Overall, I find the quality of the reviewed manuscript suitable for publication in the Cells Journal.

Line 52 – thoid -> thyroid?

Author Response

Comment 1: Due to many corrections, I strongly recommend the authors double-check if the figure panels match their descriptions in the legends and the results. For example, in the response to my question 2, the authors mention adding quantification to Fig S3C. This is also mentioned in the S3 legend (S3D) but does not appear on the figure itself. In the response, the authors also mention quantification for Figure 4C, but this panel was actually removed from the updated Figure 4. Such inconsistencies made reviewing more difficult.

Response 1: Thanks for your comments. Sorry for these errors, this time we double checked the manuscript and supplementary figures, and now we're sure all the figure panels match their descriptions in the legends and the results.

Comment 2: Line 52 – thoid -> thyroid?

Response 2: Thank you for pointing this out. We have replaced "thoid" with "thyroid". -line 47.